# Adversarial Training on Purification (AToP): Advancing Both Robustness and Generalization

**Guang Lin**[1,2], **Chao Li**[2], **Jianhai Zhang**[3], **Toshihisa Tanaka**[1,2] and **Qibin Zhao**[2,1,*]

[1]Tokyo University of Agriculture and Technology
[2]RIKEN Center for Advanced Intelligence Project (RIKEN AIP)
[3]Hangzhou Dianzi University

## Abstract

The deep neural networks are known to be vulnerable to well-designed adversarial attacks. The most successful defense technique based on adversarial training (AT) can achieve optimal robustness against particular attacks but cannot generalize well to unseen attacks. Another effective defense technique based on adversarial purification (AP) can enhance generalization but cannot achieve optimal robustness. Meanwhile, both methods share one common limitation on the degraded standard accuracy. To mitigate these issues, we propose a novel pipeline to acquire the robust purifier model, named Adversarial Training on Purification (AToP), which comprises two components: perturbation destruction by random transforms (RT) and purifier model fine-tuned (FT) by adversarial loss. RT is essential to avoid overlearning to known attacks, resulting in the robustness generalization to unseen attacks, and FT is essential for the improvement of robustness. To evaluate our method in an efficient and scalable way, we conduct extensive experiments on CIFAR-10, CIFAR-100, and ImageNette to demonstrate that our method achieves optimal robustness and exhibits generalization ability against unseen attacks.

## 1 Introduction

Deep neural networks (DNNs) have been shown to be susceptible to adversarial examples (Szegedy et al., 2014; Deng et al., 2020), which are generated by adding small, human-imperceptible perturbations to natural images, but completely change the prediction results to DNNs, leading to disastrous implications (Goodfellow et al., 2015). Since then, numerous methods have been proposed to defend against adversarial examples. Notably, adversarial training (AT) has gained significant attention as a robust learning algorithm against adversarial attacks (Goodfellow et al., 2015; Madry et al., 2018a). AT has proven to be effective in achieving state-of-the-art robustness against known attacks, albeit with a tendency to overfit the specific adversarial examples seen during training. Consequently, within the framework of AT, the persistent conundrum revolves around striking a balance between standard accuracy and robustness (Zhang et al., 2019), a dilemma that has been a subject of enduring concern. Furthermore, the robustness of AT defense against multiple or even unforeseen attacks still remains challenging (Poursaeed et al., 2021; Laidlaw et al., 2021; Tack et al., 2022).

More recently, the concept of adversarial purification (AP) has emerged, aiming to eliminate perturbations from potentially attacked images before inputting them into the classifier model (Yang et al., 2019; Shi et al., 2021; Nie et al., 2022). In contrast to AT methods, AP operates as a pre-processing step that can defend against unseen attacks without retraining the classifier model. However, AP exhibits lower robustness against known attacks compared to AT methods. Meanwhile, both AT and AP have a common limitation on the degraded standard accuracy (Tramer et al., 2020; Chen et al., 2022), which restricts their applicability in real-world applications. As summarized in Table 1, the present landscape of defense methods is confronted with a formidable challenge: *How can we*

---

* Corresponding Author

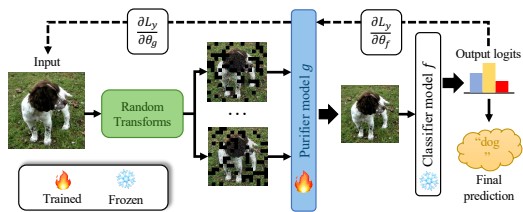

Figure 1: Illustration of adversarial training on purification (AToP).

Table 1: Robustness comparison of defenses with expectation (negative impacts are marked in red).

| Defense method | Clean images | Known attacks | Unseen attacks |
|---|---|---|---|
| Expectation | = | ↑↑ | ↑ |
| AT | ↓↓ | ↑↑ | ≈ |
| AP | ↓ | ↑ | ↑ |
| AToP (ours) | ≈ | ↑↑ | ↑ |

*enhance robustness against known attacks while maintaining generalization to unseen attacks and preserving standard accuracy on clean examples?*

To tackle these challenges with the framework of AT and AP, we propose a novel defense technique that conceptually separates the AP method into two components: perturbation destruction by random transforms and fine-tuning the purifier model via supervised AT called Adversarial Training on Purification (AToP), as shown in Figure 1. Specifically, we develop three types of random transforms and conduct a comprehensive investigation into their effectiveness in augmenting robustness and generalization against attacks. The purifier model is designed to reconstruct clean examples from corrupted inputs, which can also ensure accurate classification outputs regardless of whether the inputs are corrupted adversarial examples or corrupted clean examples. Furthermore, more importantly, we introduce the adversarial loss calculated from the output of the classifier model to fine-tune the purifier model in a supervised manner while keeping the classifier model fixed. Differing from the traditional AP methods, our method can learn a robust purifier model to generate high-quality purified examples and avoid generating examples to incorrect classes.

We empirically evaluate the performance of our method by comparing with the latest AT and AP methods across various attacks (i.e., FGSM (Goodfellow et al., 2015), PGD (Madry et al., 2018b), CW (Carlini & Wagner, 2017), AutoAttack (Croce & Hein, 2020), and StAdv (Xiao et al., 2018)) and conduct extensive experiments on CIFAR-10, CIFAR-100 and ImageNette employing multiple classifier models and purifier models. The experimental results demonstrate that our method achieves state-of-the-art results and exhibits robust generalization against unseen attacks. Furthermore, our method significantly improves the performance of the purifier model in robust classification, surpassing the performance of the same model used in recent studies (Ughini et al., 2022; Wu et al., 2023). To provide a more comprehensive insight into our results, we conduct visual comparisons among multiple examples. In summary, the contributions of this work are as follows.

- We propose adversarial training on purification (AToP), a novel defense technique that effectively combines the strengths of AT and AP to acquire the robust purifier model.

- We separate the adversarial purification process into two components, including perturbation destruction and fine-tuning purifier model, and investigate their impacts on robustness.

- We conduct extensive experiments to empirically demonstrate that the proposed method significantly improves the performance of the purifier model in robust classification.

## 2 RELATED WORK

**Adversarial training** is initially proposed by Goodfellow et al. (2015) to defend against adversarial attacks. While adversarial training has been demonstrated as an effective defense against attacks, it remains susceptible to unseen attacks (Stutz et al., 2020; Poursaeed et al., 2021; Laidlaw et al., 2021; Tack et al., 2022). Additionally, retraining the classifier model leads to substantial training expenses and could severely impair the standard accuracy of the model (Madry et al., 2018a; Tramèr et al., 2018; Zhang et al., 2019). In a study conducted by Raff et al. (2019), the impact of random transforms on the robustness of defense method was investigated. Their research revealed that the inclusion of a broader range of random transforms applied to images before adversarial training can significantly improve robustness. However, the utilization of random transforms also led to the loss of semantic information, which can severely affect the accuracy. Although our framework uses

random transforms and AT strategy, the essential difference with these works is that we aim to train a robust purifier model to recover semantics from corrupted examples using AT, rather than training the classifier model to learn the perturbation features using AT. Yang et al. (2019) also proposed a method that combines adversarial training and purification. Specifically, a purifier model was used prior to the classifier model, and AT was only applied to the classifier. Meanwhile, Ryu & Choi (2022) applied a similar pipeline but fine-tuned both the classifier model and the purifier model using AT. Consequently, the training process became time-intensive. In contrast, our method applies the adversarial loss to train the purifier model, which avoids the need for retraining the classifier model and mitigating the risk of overfitting specific attacks.

**Adversarial purification** aims to purify adversarial examples before classification, which has emerged as a promising defense method (Shi et al., 2021; Srinivasan et al., 2021). Currently, the most prevalent pipeline within AP involves the utilization of a generative model to recover purified examples from adversarial examples. Some prior studies (Bakhti et al., 2019; Hwang et al., 2019) employed consistency losses to train the purifier model, which obtains similar results to AT methods. Nevertheless, these methods primarily capture the distribution of specific perturbations, which also limits their effectiveness in defending against unseen attacks. Other studies (Nie et al., 2022; Ughini et al., 2022; Wu et al., 2023) employed the pre-trained generative model as a purifier model. These methods have acquired the ability to recover information from noisy images through extensive exposure to clean examples, enabling them to provide an effective defense against even unseen attacks. However, there is a limitation of the existing pre-trained generator-based purifier model, where all parameters are fixed and cannot be further improved for known attacks (Dai et al., 2022). To address these issues, we employ a pre-trained generative model as an initial purifier model, which can be subsequently fine-tuned using examples in a supervised manner, i.e., adversarial training on purifier model. Additionally, we introduce an adversarial loss derived from classifier outputs rather than consistency loss derived from purifier outputs. As shown in the experiments, our method achieves improved robustnessness without compromising generalization against unseen attacks.

## 3 METHOD

In this section, we introduce a novel defense pipeline to acquire the robust purifier model, named Adversarial Training on Purification (AToP), which comprises two components: random transform (RT) and fine-tuning (FT) the purifier model. Specifically, RT can significantly destruct adversarial perturbations, irrespective of the type of attack, thereby achieving effective defense against even unseen attacks. On the other hand, FT can generate purified examples from corrupted inputs with true label information, thus further enhancing robustness.

The entire classification system comprises three processes: *transform* denoted $t(\cdot)$, *purifier* denoted $g(\cdot)$, and *classifier* denoted $f(\cdot)$. Given an input $x$, the pipeline of output $y$ can be formulated as

$$x_t = t\left(x; \theta_t\right), \hat{x} = g\left(x_t; \theta_g\right), y = f\left(\hat{x}; \theta_f\right),\tag{1}$$

where $\theta_t$, $\theta_g$, $\theta_f$ represent parameters involved in the transform, purifier, and classifier, respectively.

### 3.1 DESTRUCT PERTURBATION STRUCTURE BY RANDOM TRANSFORMS

We utilize three types of random transforms, ranging from simple to complex cases.

**The first type of transform** ($\text{RT}_1$) utilizes a binary mask $m$ with small patches of size $p \times p$ that are randomly missing with missing rate $r$ (Yang et al., 2019). As shown in Eq. 2, given any input $x$, we take the element-wise multiplication $m \odot x$,

$$x_t = m \odot x, \quad \hat{x} = g(x_t).\tag{2}$$

**The second type of transform** ($\text{RT}_2$) involves the addition of Gaussian noise $\eta$ before applying the random mask $m$ to further destruct the perturbations. As shown in Figure 2(a), we have

$$x_t = m \odot (x + \eta), \quad \hat{x} = g(x_t).\tag{3}$$

Randomized smoothing by adding Gaussian noise is a method that provides provable defenses (Li et al., 2019; Cohen et al., 2019) where Cohen et al. (2019) theoretically provides robustness guarantees

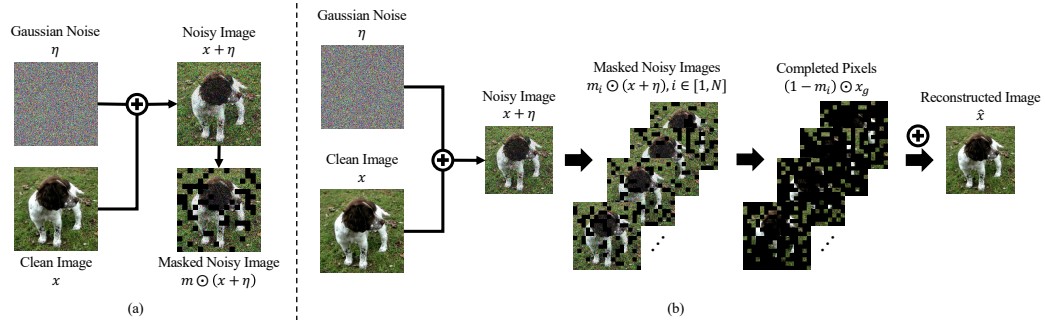

Figure 2: Illustration of random transforms. (a) Firstly, adding Gaussian noise to preliminarily corrupt the image, then randomly masking the image. (b) Next, based on (a), the noisy image is covered randomly by $N$ non-overlapping masks. Finally, the completed pixels are combined to reconstruct the image denoted as $\hat{x}$.

for classifiers. In the latest work, Wu et al. (2023) enhance robustness by learning representations through a generative model, as shown in Eq. 3.

**The third type of transform** ($\text{RT}_3$) involves the repetition of $N$ transformations on a single image to ensure that all pixel values are regenerated by the purifier model, thereby removing as many perturbations as possible (Ughini et al., 2022). As shown in Figure 2(b), the noisy image is randomly covered by $N$ non-overlapping masks $m_i$. After being processed by purifier model, only the pixels generated by the purifier model are aggregated to yield $\hat{x}$, which is

$$x_t^i = m_i \odot (x + \eta), \quad x_g^i = g(x_t^i), \quad \hat{x} = \sum_{i=1}^{N} (1 - m_i) \odot x_g^i. \tag{4}$$

## 3.2 FINE-TUNING THE PURIFIER MODEL WITH ADVERSARIAL LOSS

---

**Algorithm 1** Adversarial Training on Purification method (AToP)

---

**Require:** Training examples $x$, ground truth $y$, parameters of classifier model $\theta_f$, parameters of purifier model $\theta_g$, training epoch $N_{ep}$
1: Initialize $\theta_f$ and $\theta_g$ with pre-trained classifier model and pre-trained purifier model.
2: **for** epoch = $1...N_{ep}$ **do**
3:     Build adversarial examples $x'$ with perturbations $\delta$:    $x' \leftarrow x + \delta$
4:     Freeze $\theta_f$ and update $\theta_g$ with gradient descent based on loss in Eq. (9).
5:     $\theta_g \leftarrow \theta_g - \nabla \theta_g$
6: **end for**
7: **return** purifier model with $\theta_g$

---

Having described the random transforms, we proceed to provide a description of the purifier model and fine-tuning process. Specifically, based on the pre-trained generator-based purifier model trained by the original loss function $L_{org}$, we propose AToP loss function $L_{\theta_g}$, incorporating an adversarial loss to fine-tune the purifier model,

$$L_{\theta_g} = L_{org}(x, \theta_g) + \lambda \cdot L_{cls}(x, y, \theta_g, \theta_f), \tag{5}$$

where $\lambda$ is held constant as hyperparameter. In the paper, we improve upon two recent studies, and the original loss function $L_{org}$ is denoted as $L_{df}$ in the GAN-based model (Ughini et al., 2022) and as $L_{mae}$ in the AE-based model (Wu et al., 2023). For instance, we utilize the GAN-based model called DeepFill as the purifier model $g$, in conjunction with $\text{RT}_3$ and the classifier model F. Given any input $x$, our defense process is defined as below:

$$x_t^i = t(x; \theta_t) = \text{RT}_3(x) = m_i \odot (x + \eta), \tag{6}$$

$$x_g^i = g(x_t^i; \theta_g) = \text{DeepFill}(x_t^i), \quad \hat{x} = \sum_{i=1}^{N} (1 - m_i) \odot x_g^i, \tag{7}$$

$$y = f(\hat{x}; \theta_f) = \text{F}(\hat{x}), \tag{8}$$

where $\theta_t$ contains three components: Gaussian standard $\sigma$, missing rate $r$, and mask number $N$. The parameter $\theta_f$ is derived from the pre-trained classifier model. To perform robust fine-tuning of the purifier model using adversarial training, we aim to optimize the model by freezing classifier parameter $\theta_f$ and updating purifier parameter $\theta_g$, where $\theta_f$ and $\theta_g$ are initialized by the pre-trained classifier model and the pre-trained generative model, respectively, as shown in Algorithm 1.

For pre-training GAN-based model (Ughini et al., 2022), we have $L_{org} = L_{df}(x, \theta_g) = \mathbb{E}[D(x)] - \mathbb{E}[D(g(t(x), \theta_g))] + \mathbb{E}[\|x - g(t(x), \theta_g)\|_{\ell_1}]$. To further optimize $\theta_g$, we propose AToP loss function and fine-tune the pre-trained GAN-based model, which comprises an original loss $L_{df}$, as well as an additional adversarial loss $L_{cls}$. Given an adversarial example $x'$ with perturbation $\delta$, we have

$$
\begin{aligned}
L_{\theta_g} &= L_{df}(x', \theta_g) + \lambda \cdot L_{cls}(x', y, \theta_g, \theta_f) \\
&= \mathbb{E}[D(x')] - \mathbb{E}[D(g(t(x'), \theta_g))] + \mathbb{E}[\|x' - g(t(x'), \theta_g)\|_{\ell_1}] \\
&\quad + \lambda \cdot \max_{\delta} CE\{y, \mathrm{F}[g(t(x'), \theta_g)]\}, \text{ where } x' = x + \delta.
\end{aligned}
\tag{9}
$$

Here, $L_{df}$ comprises the first three terms, where $D$ represents the discriminator, and $g$ is the generator-based purifier model, which plays similar roles as the generative adversarial network model. During training, the discriminator $D$ is responsible for distinguishing between real examples and the purified examples $g(t(x), \theta_g)$. Simultaneously, the purifier model $g$ generates high-quality examples to deceive the discriminator $D$. The third term is consistency loss $\|x - g(t(x), \theta_g)\|_{\ell_1}$, which encourages strong image restoration ability to the original examples $x$. Consequently, $L_{df}$ effectively ensures that the purified examples are not only high-quality but also closely resemble their original example. If RT is sufficiently strong to the extent that $t(x)$ is essentially Gaussian noise, $L_{df}$ loss can ensure the purified examples are generated from the same distribution of $x$ and are of high-quality. Conversely, if RT is minimal such as $t(x) = x$, $L_{df}$ is instrumental in guaranteeing that the purifier model can achieve an exact recovery of the original examples. Consequently, if RT is strong enough to destruct the adversarial perturbations, the model can guarantee to generate examples from the original distribution while avoiding the reconstruction of adversarial perturbations. Thus, the effectiveness remains unaffected even for unseen attacks.

During fine-tuning, $L_{cls}$ corresponds to the last term, which serves the purpose of further ensuring that our purified examples $\hat{x}$ do not contain sufficient adversarial perturbations that fool the classifier. On the other hand, if RT is very strong and purified examples $\hat{x}$ do not contain any adversarial perturbations, $L_{cls}$ can also ensure $\hat{x}$ will not be high-quality examples drawn from different classes. Intuitively, the adversarial perturbations $\delta$ are generated by maximizing classification loss, which can be used as the new examples to train our purifier model such that adversarial perturbations can be reconstructed up to the degree that remains correct classification. Unlike standard AT, our adversarial loss is not used to update the classifier model but to optimize the purifier model. As compared to existing AP methods using adversarial examples (Bakhti et al., 2019; Hwang et al., 2019), our adversarial loss is essentially more robust than either consistency loss or generator-style loss. Finally, $\lambda > 0$ represents a weight hyperparameter to balance the generative error and classification error.

## 4 EXPERIMENTS

In this section, we conduct extensive experiments on CIFAR-10, CIFAR-100 and ImageNette across various transforms, classifier models and purifier models on attack benchmarks. Our findings demonstrate that the proposed method achieves state-of-the-art results and exhibits generalization ability against unseen attacks. Specifically, compared to against AutoAttack $l_\infty$ on CIFAR-10, our method improves the robust accuracy by 13.28% on GAN-based purifier model and 14.45% on AE-based purifier model, respectively. Furthermore, we empirically study the random transforms and discuss their impacts on robust accuracy. Finally, we provide visual examples (more details can be found in supplementary materials) to provide a more comprehensive understanding of our method.

### 4.1 EXPERIMENTAL SETUP

**Datasets and model architectures:** We conduct extensive experiments on CIFAR-10, CIFAR-100 (Krizhevsky et al., 2009) and ImageNette (a subset of 10 classified classes from ImageNet) (Howard, 2021) to empirically validate the effectiveness of the proposed methods against adversarial attacks. For the classifier model, we utilize the pre-trained ResNet model (He et al., 2016) and WideResNet

model (Zagoruyko & Komodakis, 2016). For the purifier model, we utilize the pre-trained GAN-based model (Ughini et al., 2022) and AE-based model (Wu et al., 2023) and utilize adversarial examples generated by FGSM for the fine-tuning.

**Adversarial attacks:** We evaluate our method against various attacks: We utilize AutoAttack $l_\infty$ and $l_2$ threat models (Croce & Hein, 2020) as one benchmark, which is a powerful attack that combines both white-box and black-box attacks, and utilizes Expectation Over Time (EOT) (Athalye et al., 2018b) to tackle the stochasticity introduced by random transforms. To consider unseen attacks without $l_p$-norm, we utilize spatially transformed adversarial examples (StAdv) (Xiao et al., 2018) for validation. Additionally, we generate adversarial examples using some standard methods, including Fast Gradient Sign Method (FGSM) (Goodfellow et al., 2015), Projected Gradient Descent (PGD) (Madry et al., 2018b) and Carlini-Wagner (CW) (Carlini & Wagner, 2017) attacks.

**Evaluation metrics:** We evaluate the performance of defense methods using two metrics: standard accuracy and robust accuracy, obtained by testing on clean examples and adversarial examples, respectively. Due to the high computational cost of testing models with multiple attacks, following guidance by Nie et al. (2022), we randomly select 512 images from the test set for robust evaluation.

**Training details:** After experimental testing, we have determined the hyperparameters: Gaussian standard deviation $\sigma = 0.25$, mask size $p = P/8$ ($P$ is side length of input $x$), mask number $N = 4$, missing rate $r = 0.25$ and weight $\lambda = 0.1$. All experiments presented in the paper are conducted

Table 2: Standard accuracy and robust accuracy against AutoAttack $l_\infty$ threat ($\epsilon = 8/255$) on CIFAR-10 with WideResNet classifier model.

| Defense method | Extra data | Standard Acc. | Robust Acc. |
|---|---|---|---|
| WideResNet-28-10 | | | |
| Zhang et al. (2020) | ✓ | 85.36 | 59.96 |
| Gowal et al. (2020) | ✓ | 89.48 | 62.70 |
| Rebuffi et al. (2021) | × | 87.33 | 61.72 |
| Gowal et al. (2021) | × | 87.50 | 65.24 |
| Nie et al. (2022) | × | 89.02 | 70.64 |
| Ours | × | **90.62** | **72.85** |
| WideResNet-70-16 | | | |
| Gowal et al. (2020) | ✓ | 91.10 | 65.87 |
| Rebuffi et al. (2021) | ✓ | 92.23 | 66.56 |
| Rebuffi et al. (2021) | × | 88.54 | 64.46 |
| Gowal et al. (2021) | × | 88.74 | 66.60 |
| Nie et al. (2022) | × | 90.07 | 71.29 |
| Ours | × | **91.99** | **76.37** |

Table 3: Standard accuracy and robust accuracy against AutoAttack $l_2$ threat ($\epsilon = 0.5$) on CIFAR-10 with WideResNet classifier model.

| Defense method | Extra data | Standard Acc. | Robust Acc. |
|---|---|---|---|
| WideResNet-28-10 | | | |
| Augustin et al. (2020) | ✓ | 92.23 | 77.93 |
| Ding et al. (2019) | × | 88.02 | 67.77 |
| Rebuffi et al. (2021) | × | **91.79** | 78.32 |
| Nie et al. (2022) | × | 91.03 | 78.58 |
| Ours | × | 90.62 | **80.47** |
| WideResNet-70-16 | | | |
| Rebuffi et al. (2021) | ✓ | 95.74 | 81.44 |
| Gowal et al. (2020) | × | 90.90 | 74.03 |
| Nie et al. (2022) | × | **92.68** | 80.60 |
| Ours | × | 91.99 | **81.35** |

Table 4: Standard accuracy and robust accuracy against AutoAttack $l_\infty$ threat ($\epsilon = 8/255$) on CIFAR-100 with WideResNet classifier model.

Table 5: Robust accuracy against AutoAttack $l_\infty$ ($\epsilon = 8/255$) on CIFAR-10 and CIFAR-100,AutoAttack $l_2$ ($\epsilon = 0.5$) on CIFAR-10 with WideResNet-28-10 classifier model.

| Defense method | CIFAR 10, $l_\infty$ | CIFAR 10, $l_2$ | CIFAR 100, $l_\infty$ |
|---|---|---|---|
| Wu et al. (2023) | 44.73 | 45.12 | 38.48 |
| Wu et al. (2023) +AToP (Ours) | **59.18** | **65.43** | **40.23** |
| Ughini et al. (2022) | 59.57 | 63.69 | 38.89 |
| Ughini et al. (2022) +AToP (Ours) | **72.85** | **80.47** | **43.55** |

| Defense method | Extra data | Standard Acc. | Robust Acc. |
|---|---|---|---|
| WideResNet-28-10 | | | |
| Hendrycks et al. (2019) | ✓ | 59.23 | 28.42 |
| Pang et al. (2022) | × | 63.66 | 31.08 |
| Cui et al. (2023) | × | 73.85 | 39.18 |
| Ours | × | **74.61** | **43.55** |
| WideResNet-70-16 | | | |
| Bai et al. (2023) | ✓ | 85.21 | 38.72 |
| Wang et al. (2023) | × | 75.22 | 42.67 |
| Ours | × | **76.95** | **44.44** |

under these hyperparameters and performed by NVIDIA RTX A5000 Graphics Card with 24GB GDDR6 GPU memory, CUDA v11.7 and cuDNN v8.5.0 in PyTorch v1.13.11 (Paszke et al., 2019).

## 4.2 COMPARISON WITH STATE-OF-THE-ART METHODS

We evaluate our method against AutoAttack $l_\infty$ and $l_2$ threat models and compare its robustness to state-of-the-art methods listed in RobustBench (Croce et al., 2021). All experiments presented in this section utilize $\mathrm{RT}_2$ and GAN-based model, which yielded the best results in the experiment.

**Result analysis on CIFAR-10:** Table 2 shows the performance of the defense methods against AutoAttack $l_\infty$ ($\epsilon = 8/255$) threat model on CIFAR-10 with WideResNet-28-10 and WideResNet-70-16 classifier models. Our method outperforms all other methods without extra data (the dataset introduced by Carmon et al. (2019)) in terms of both standard accuracy and robust accuracy. Specifically, compared to the second-best method, our method improves the robust accuracy by 2.21% on WideResNet-28-10 and by 5.08% on WideResNet-70-16, respectively. Furthermore, in terms of robust accuracy, our method even outperforms the methods with extra data. Table 3 shows the performance of the defense methods against AutoAttack $l_2$ ($\epsilon = 0.5$) threat model. Our method outperforms all other methods without extra data in terms of robust accuracy. Specifically, compared to the second-best method, our method improves the robust accuracy by 1.89% on WideResNet-28-10 and by 0.75% on WideResNet-70-16, respectively.

Table 5 shows the performance of the two generator-based purifier models (Ughini et al., 2022; Wu et al., 2023) against AutoAttack $l_\infty$ ($\epsilon = 8/255$) and $l_2$ ($\epsilon = 0.5$) threat models with WideResNet-28-10 classifier model. Specifically, our method improves the robust accuracy by 14.45% and 20.31% on AE-based purifier model, 13.28% and 16.87% on GAN-based purifier model, respectively. The results demonstrate that our method can significantly improve the robustness of the purifier model.

**Result analysis on CIFAR-100:** Table 4 shows the performance of the defense methods against AutoAttack $l_\infty$ ($\epsilon = 8/255$) threat model on CIFAR-100 with WideResNet-28-10 and WideResNet-70-16 classifier models. Our method outperforms all other methods without extra data in terms of both standard accuracy and robust accuracy. Specifically, compared to the second-best method, our method improves the robust accuracy by 4.37% on WideResNet-28-10 and by 1.77% on WideResNet-70-16, respectively. Table 5 shows the performance of the models against AutoAttack $l_\infty$ ($\epsilon = 8/255$) and the observations are basically consistent with CIFAR-10, demonstrating the effectiveness of our method in enhancing robust classification across different datasets.

## 4.3 DEFEND AGAINST UNSEEN ATTACKS

As previously mentioned, the mainstream adversarial training (AT) methods have shown limitations in defending against unseen attacks (Laidlaw et al., 2021; Dolatabadi et al., 2022), even can only defend against known attacks. To demonstrate the generalization ability of our method with $\mathrm{RT}_2$ and GAN, we conduct experiments in which defense methods are evaluated against attacks with

Table 6: Standard accuracy and robust accuracy against AutoAttack $l_\infty$ ($\epsilon = 8/255$), $l_2$ ($\epsilon = 1$) and StAdv non-$l_p$ ($\epsilon = 0.05$) threat models on CIFAR-10 with ResNet-50 classifier model. We utilize GAN-based model with $\mathrm{RT}_2$, and all settings follow the same as Laidlaw et al. (2021).

| Defense method | Standard Acc. | $l_\infty$ | $l_2$ | StAdv |
|---|---|---|---|---|
| Standard Training | 94.8 | 0.0 | 0.0 | 0.0 |
| Adv. Training with $l_\infty$ (Laidlaw et al., 2021) | 86.8 | 49.0 | 19.2 | 4.8 |
| Adv. Training with $l_2$ (Laidlaw et al., 2021) | 85.0 | 39.5 | 47.8 | 7.8 |
| Adv. Training with StAdv (Laidlaw et al., 2021) | 86.2 | 0.1 | 0.2 | 53.9 |
| Adv. Training with all (Laidlaw et al., 2021) | 84.0 | 25.7 | 30.5 | 40.0 |
| PAT-self (Laidlaw et al., 2021) | 82.4 | 30.2 | 34.9 | 46.4 |
| Adv. CRAIG (Dolatabadi et al., 2022) | 83.2 | 40.0 | 33.9 | 49.6 |
| DiffPure (Nie et al., 2022) | 88.2 | 70.0 | 70.9 | 55.0 |
| Ours | **89.1** | **71.2** | **73.4** | **56.4** |

varying constraints (including AutoAttack $l_\infty$, $l_2$ and StAdv non-$l_p$ threat models) on CIFAR-10 with ResNet-50 classifier model.

Table 6 shows that AT method can achieve robustness against particular attacks seen during training (as indicated by Acc. with underscore) but struggles to generalize robustness to unseen attacks. The intuitive idea is to apply AT across all different attacks to obtain a robust model. However, it is challenging to train such a model due to the inherent differences among all sorts of attacks. Furthermore, the emergence of new attack techniques continually makes it impractical to train the model to defend against all attacks. We combine AT and AP as a new defense method, which outperforms all other methods in terms of both standard accuracy and robust accuracy. Specifically, compared to the second-best method, our method improves the robust accuracy by 1.2%, 2.5% and 1.4% on AutoAttack $l_\infty$, $l_2$ and StAdv non-$l_p$, respectively. The results demonstrate that our method achieves state-of-the-art results and exhibits generalization ability against unseen attacks.

## 4.4 ABLATION STUDIES

We conduct ablation studies to validate the effectiveness of factors within our method in defending against more standard attacks. Considering the robustness misestimation caused by obfuscated gradients of purifier model, we use BPDA (Athalye et al., 2018a) following the setting by Yang et al. (2019), which approximates the gradient of purifier model as 1 during backward pass. Additionally, we use EOT (Athalye et al., 2018b) following the setting by Lee & Kim (2023) with 20 iterations.

Table 7: Standard accuracy and robust accuracy of attacking the classifier model on CIFAR-10 with ResNet-18. All attacks are $l_\infty$ threat model with $\epsilon = 8/255$.

| Transforms | AToP | Standard Acc. | FGSM | PGD-10 | PGD-20 | PGD-1000 | CW-100 |
|---|---|---|---|---|---|---|---|
| RT$_1$ | × | **93.36** | 16.60 | 0.00 | 0.00 | 0.00 | 21.09 |
| | ✓ | **93.36** | **91.99** | **43.55** | **36.72** | **39.45** | **87.89** |
| RT$_2$ | × | 84.18 | 55.08 | 72.27 | 70.90 | 67.97 | 87.70 |
| | ✓ | **90.04** | **89.84** | **84.77** | **84.57** | **84.38** | **88.87** |
| RT$_3$ | × | 75.98 | 67.97 | 70.51 | 70.70 | 70.31 | 75.20 |
| | ✓ | **80.02** | **70.90** | **73.05** | **72.07** | **73.44** | **76.56** |

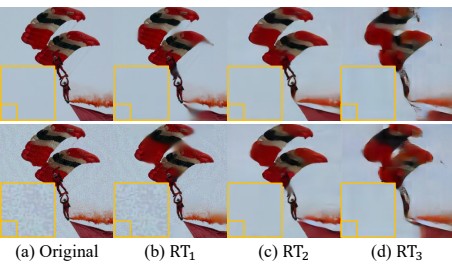

(a) Original    (b) RT$_1$    (c) RT$_2$    (d) RT$_3$

Figure 3: The purified images are obtained for clean (Top) and adversarial examples (Bottom) with different random transforms.

Table 8: Standard accuracy and robust accuracy against PGD+EOT $l_2$ threat ($\epsilon = 1.0$) on CIFAR-10 with ResNet-18 classifier model.

| | AToP | Standard | PGD-10 | PGD-20 |
|---|---|---|---|---|
| RT$_1$ | × | **93.36** | 20.31 | 15.04 |
| | ✓ | **93.36** | **27.34** | **22.46** |
| RT$_2$ | × | 84.18 | 67.19 | 63.09 |
| | ✓ | **90.04** | **69.92** | **68.75** |
| RT$_3$ | × | 75.98 | 28.52 | 26.17 |
| | ✓ | **80.02** | **46.48** | **41.60** |

**Result analysis on CIFAR-10:** Table 7 shows the performance of the defense model utilizing different transforms against FGSM, PGD and CW on CIFAR-10 with ResNet-18 classifier model. The numbers after the attack represent the number of steps. For each transform, our method significantly outperforms the method without AToP in terms of both standard accuracy and robust accuracy.

Figure 4a shows the standard accuracy and robust accuracy of the purifier model trained with clean examples and adversarial examples, respectively. The results demonstrate that while utilizing clean examples can mitigate the impact of semantic loss induced by random transforms, thereby improving the standard accuracy. Moreover, utilizing adversarial examples can improve the robust accuracy. Specifically, AToP with clean examples achieves the best standard accuracy, while AToP with

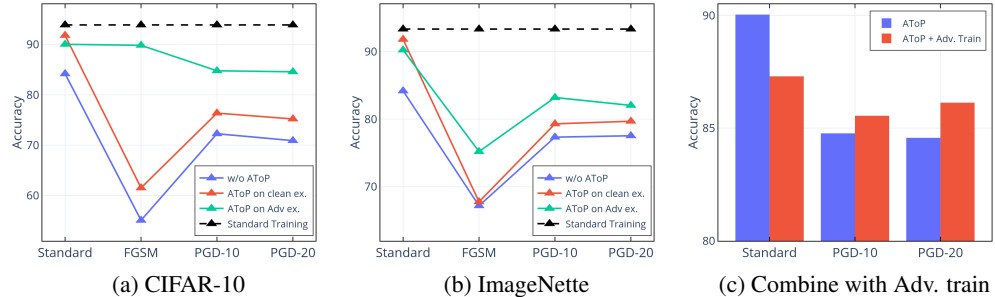

(a) CIFAR-10      (b) ImageNette      (c) Combine with Adv. train

Figure 4: Standard accuracy and robust accuracy of the GAN-based purifier model trained with clean examples and adversarial examples, respectively, on (a) CIFAR-10 and (b) ImageNette. The dashed line represents the accuracy of standard training w/o attacks. (c) Standard accuracy and robust accuracy of our method with adversarially trained ResNet-18 on CIFAR-10.

Table 9: Standard accuracy and robust accuracy of attacking the classifier model on ImageNette with ResNet-34. All attacks are $l_\infty$ threat model with $\epsilon = 8/255$.

| Transforms | AToP | Standard Acc. | FGSM | PGD-10 | PGD-20 | PGD-1000 | CW-100 |
|---|---|---|---|---|---|---|---|
| $\text{RT}_1$ | $\times$ | **91.99** | 18.95 | 0.00 | 0.00 | 0.00 | 64.26 |
| | $\checkmark$ | 89.84 | **40.62** | **45.31** | **45.12** | **47.07** | **78.12** |
| $\text{RT}_2$ | $\times$ | 84.18 | 67.19 | 77.34 | 77.54 | 77.34 | 86.52 |
| | $\checkmark$ | **90.23** | **75.20** | **83.20** | **82.03** | **82.03** | **88.48** |
| $\text{RT}_3$ | $\times$ | 71.48 | 66.60 | 67.77 | 64.26 | 66.21 | 67.38 |
| | $\checkmark$ | **80.08** | **75.78** | **76.37** | **75.98** | **72.85** | **78.32** |

adversarial examples achieves the best robustness. Additionally, AP is a pre-processing technique that can combine with an adversarial trained classifier to further enhance robust accuracy, as shown in Figure 4c. However, the improvement comes at the cost of a reduction in standard accuracy.

For the more comprehensive evaluation, we follow the guidance of Lee & Kim (2023) to evaluate our method against PGD+EOT with $l_2$ threat ($\epsilon = 1.0$). Table 8 shows the performance of the defense model with different transforms, and our method still significantly outperforms the method without AToP in terms of both standard accuracy and robust accuracy, demonstrating the effectiveness of our method in enhancing the performance of the purifier model in robust classification.

**Result analysis on ImageNette:** Table 9 shows the performance of the defense model with different transforms against FGSM, PGD and CW on ImageNette with ResNet-34 classifier model. Figure 4b shows the standard accuracy and robust accuracy of the purifier model trained with clean examples and adversarial examples, respectively. The experimental observations on ImageNette are basically consistent with CIFAR-10, with our method significantly outperforming the method without AToP in terms of robust accuracy. On the other hand, we can observe that while employing more powerful random transforms within the same AToP setting, the perturbations will be gradually destructed, resulting in an increase in robustness; however, semantic information will also be lost, decreasing standard accuracy. This trade-off presents a limitation that prevents the continuous improvement of robust accuracy. Specifically, $\text{RT}_2$ achieves optimal robustness while maintaining a satisfactory standard accuracy. Figure 3 shows the purified images obtained by inputting clean examples and adversarial examples into purifier model with various RTs. As previously mentioned, while RTs effectively destruct perturbations, they can also introduce the loss of semantic information.

So far, we have conducted extensive experiments across various attacks, classifiers, generator-based purifiers, datasets and RT settings, which have consistently demonstrated the effectiveness of our method in acquiring the robust purifier model for enhancing robust classification.

**Limitations:** Given that our method involves the fine-tuning of the generative model, the complexity of the generative model has a direct impact on the computational cost of AToP. Initially, we explore fine-tuning the diffusion model to improve its robustness. However, as purified images need to be fed

into the classifier, generating 1000 images and fine-tuning the diffusion model requires 144 minutes. In contrast, with the same setting, the GAN-based purifier model only requires 62 seconds.

## 5 CONCLUSION

In the paper, we propose a novel defense technique called AToP that utilizes adversarial training to acquire the robust purifier model. To demonstrate the robustness of our method, we conduct extensive experiments on CIFAR-10, CIFAR-100 and ImageNette. In defense of various attacks under multiple transforms, classifier and purifier architectures, our method consistently achieves state-of-the-art results and exhibits generalization ability against unseen attacks. Ablation studies further highlight that our method can significantly improve the performance of the purifier model on robust classification. Despite the large improvements achieved, our method has a major limitation: AToP requires training on the purifier model, and as the complexity of the purifier model increases, so does the training cost. However, exploring this direction presents an intriguing research opportunity for developing a more efficient defense method by combining AT and AP.

## ACKNOWLEDGMENTS

Guang Lin was supported by the RIKEN Junior Research Associate Program. This work was supported in part by the JSPS KAKENHI Grant Numbers JP20H04249, JP23K28109, 24K03005, and RIKEN Incentive Research Project.

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

## SUPPLEMENTARY MATERIAL

### PURIFICATION PROCESS AND VISUALIZATION

First, we develop a total of three types of random transforms that employ a mixed strategy of adding Gaussian noise and missing pixels. In this section, we use images from ImageNette as examples, and all operations are based on images of size $256 \times 256$.

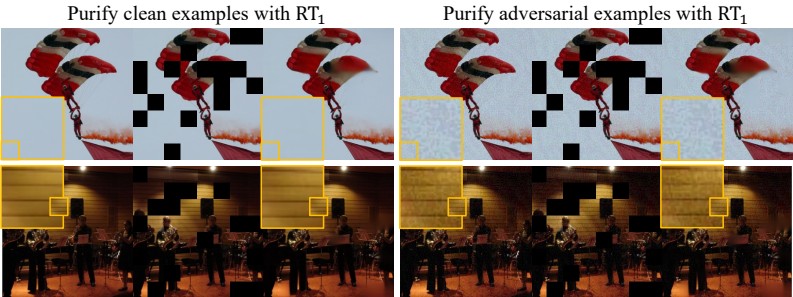

Figure 5: For each group, the first column shows the clean example (left) and adversarial example (right). The following one is the masked image. The last column illustrates the purified image.

For the first type of transform ($RT_1$), we use a binary mask with small patches of size $32 \times 32$ that are randomly missing with the missing rate $r = 0.25$. As shown in Figure 5, we illustrate the clean examples and the adversarial examples generated by FGSM attack ($\epsilon = 8/255$), which are processed by $RT_1$ and purified by the pre-trained GAN-based model without adversarial training. To facilitate image observation, we enlarge some patches of size $32 \times 32$ and mark them with yellow. In the area without masking, the perturbations are retained, leading to low robust accuracy.

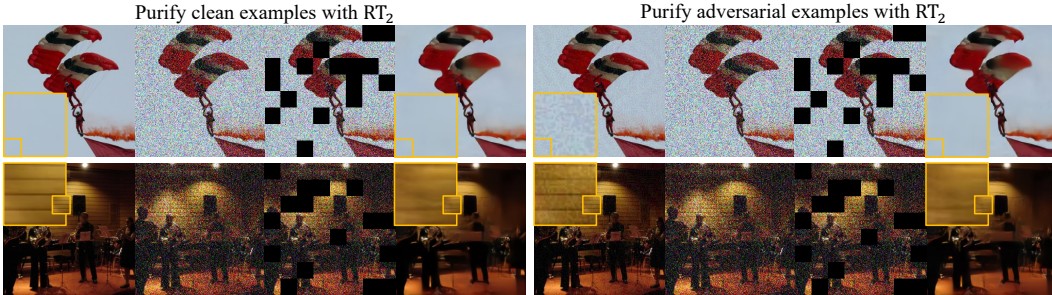

Figure 6: For each group, the first column shows the clean example (left) and adversarial example (right). The following two correspond to the noisy image with Gaussian noise and the masked noisy image. The last column illustrates the purified image.

For the second type of transform ($RT_2$), we add Gaussian noise with $\sigma = 0.25$ before applying the random mask with the missing rate $r = 0.25$ to destruct the perturbations further. As shown in Figure 6, we also illustrate the clean examples and the adversarial examples generated by FGSM attack ($\epsilon = 8/255$), which are processed by $RT_2$ and purified by the pre-trained GAN-based model without adversarial training. Since Gaussian noise can destruct some small-epsilon perturbations, therefore, in the area without masking, the perturbations are also removed, increasing robust accuracy.

For the third type of transform ($RT_3$), based on the $RT_2$, we repeat $N = 4$ transformations on the one image to ensure that all pixel values were regenerated by the purifier model to remove as much perturbations as possible. Specifically, the noisy image is covered randomly by four non-overlapping masks with the missing rate $r = 0.25$, and then we run the pre-trained GAN-based model four times by inputting masked noisy images. As shown in Figure 7, the purified image is obtained by aggregating the completed pixels of four inpainted images, where the inpainting process is carried out by the pre-trained GAN-based model without adversarial training. We compare the purified images

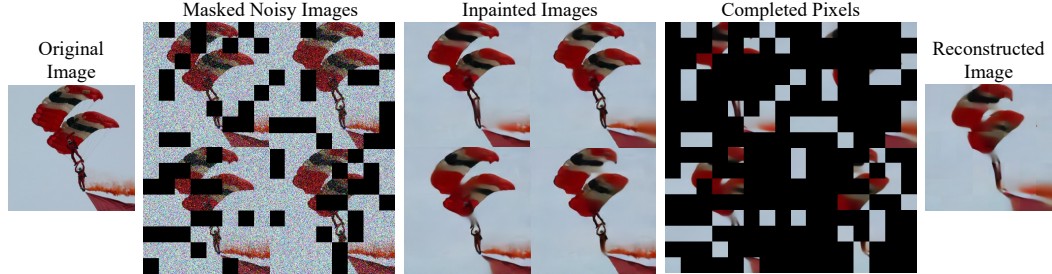

Figure 7: Purify the original image with $\mathrm{RT}_3$.

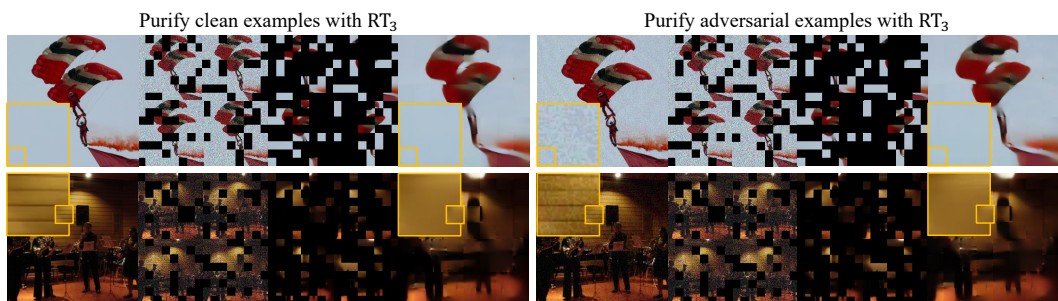

Figure 8: For each group, the first column shows the clean example (left) and adversarial example (right). The following two correspond to the masked noisy image and the completed pixels. The last column illustrates the purified image.

obtained by applying a purifier model on both the clean example and the adversarial example on the same image. As shown in Figure 8, we find that the two purified images are almost identical, which confirms that the purified image contains almost no perturbations. However, due to the excessive loss of semantic information, the standard accuracy decreased, thereby limiting the improvement of robust accuracy.

In summary, combining with Figure 5, Figure 6, and Figure 8, we can see that when using more powerful random transforms, the perturbations will be gradually destroyed; however, semantic information will also be lost. This limitation prevents the continuous increase in robust accuracy.

We fine-tune the purifier model with adversarial training. As shown in Figure 9, we illustrate purified images obtained through the three above-mentioned transforms and AToP. Observedly, we find that while suppressing the perturbations, AToP introduces some additional information to improve classification accuracy.

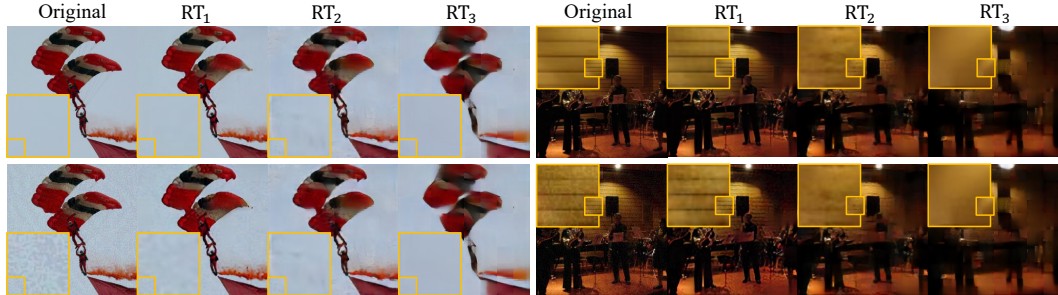

Figure 9: For each group, the first column shows the original images containing a clean example (top) and an adversarial example (bottom). The following three correspond to the images obtained with different random transforms before inputting into the purifier model.

