# OpenReview forum: "Adversarial Training on Purification (AToP): Advancing Both Robustness and Generalization"
_ICLR.cc/2024/Conference — ICLR 2024 poster_

### Official Review · Reviewer_onZy · 2023-10-24

**Soundness:** 3 good
**Presentation:** 3 good
**Contribution:** 2 fair
**Rating:** 6
**Confidence:** 4

**Summary:**

The paper introduces an adversarial defense technique called Adversarial Training on Purification (AToP). AToP conceptualizes AP into two components: perturbation destruction by random transforms and fine-tu . ning the purifier model using supervised AT. The random transforms aim to enhance the robustness to unseen attacks, while the purifier model reconstructs clean data from corrupted data, ensuring correct classification output regardless of the corruption. The purifier model is fine-tuned through an adversarial loss calculated from the classifier output. The paper presents experiments on CIFAR and ImageNette datasets, comparing AToP with other AT and AP methods. The results demonstrate that AToP achieves good generalization ability against unseen attacks, and maintains standard accuracy on clean examples.

**Strengths:**

- The paper addresses a significant problem in the field of deep learning, namely the vulnerability of neural networks to adversarial attacks.
- The proposed AToP technique offers a promising solution that combines the advantages of AT and AP, achieving good robustness, standard accuracy, and robust generalization to unseen attacks.
- The proposed AToP framework is well-defined, with a clear explanation of its components and their roles in achieving optimal robustness, standard accuracy, and robust generalization to unseen attacks.
- The experiments are conducted on CIFAR and ImageNette dataset
- I appreciate the evaluation with BDPA
- The paper is well-structured, with a clear and concise presentation of ideas.

**Weaknesses:**

- Overall the technical novelty of this work is limited. In essence, AToP combines the orthogonal techniques of AT and AP.
- The authors mainly evaluate ResNet and Wide-ResNet architecture. Hence it is not clear if this work generalizes to other model architectures.
- These days, transformer architectures are gaining popularity. How does the proposed AToP perform on such architectures, such as ViT?
- While I appreciate the evaluation on BPDA, the authors should include a separate section on “obfuscated gradients”, where they follow the guidelines of BPDA and [A] to test for obfuscated gradients.

[B] On Evaluating Adversarial Robustness; arXiv 2019

**Questions:**

Please address the points in my weaknesses section.

---

> ### Author Response · Authors · 2023-11-13
>
> We greatly appreciate the comments of the Reviewer onZy. Below are our responses to the questions raised.
>
> A1: In the related work section, we conduct a comprehensive comparison to highlight the distinctions between our method and other existing methods. Considering the limitation of the existing pre-trained generator-based purifier model, where all parameters are fixed and cannot be further improved for known attacks. Regarding this challenge, We propose a novel defense pipeline as shown in following equations.
>
> Pre-training:
>
> $L_{\theta_{g}} =L_{df}(x, \theta_{g})$
>
> $=\mathbb{E}[D(x)]-\mathbb{E}[D(g(t(x),\theta_g))]+\mathbb{E}[{\|x-g(t(x), \theta_g)\|}_{\ell_1}],$
>
> Fine-tuning:
>
> $L_{\theta_{g}} =L_{df}(x', \theta_{g})+\lambda \cdot L_{c l s}(x', y, \theta_{g}, \theta_{f})$
>
> $ =\mathbb{E}[D(x')]-\mathbb{E}[D(g(t(x'),\theta_g))]+\mathbb{E}[{\|x'-g(t(x'), \theta_g)\|}_{\ell_1}]$
>
> $ +\lambda \cdot \max_{\delta} CE(y, f(g(t(x'),\theta_g))), \quad where \ x' = x+\delta.$
>
>
>
> A2: One of the advantages of AP is that once the generator model is fully trained, it will be a plug-and-play module that can be applied to other model architectures.
>
> A3: In the experiments comparing with the state-of-the-art methods, all classifiers used are from the model zoo in RobustBench, and we select two of the most common model architectures for the experiments. To address your question, we recheck all available classifiers. However, we do not find a vanilla transformer-based classifier. There is only one transformer-based classifier with adversarial training (XCiT [1]). Therefore, we conduct experiments on XCiT, which is equivalent to AToP+AT. Following the settings of Table 2 and Table 4, here are the results:
>
> |  | CIFAR-10 | | CIFAR-100 |  |
> |:---:|:---:|:---:|:---:|:---:|
> |  | Standard Acc. | Robust Acc. | Standard Acc. | Robust Acc. |
> | XCiT-S12 [1] | 88.48 | 80.27 | 73.44 | 52.73 |
>
> [1] A Light Recipe to Train Robust Vision Transformers. 2022.
>
> A4: In Section 4.4, we conduct experiments using BPDA based on the guidance of Athalye et al. In the ablation study, we use whether to apply AToP as the only variable. This means that in a set of comparative experiments, only the training parameters of the purifier model differ, while other details remain consistent, including the parts that cause obfuscated gradients. It is reasonable that we use the paradigm provided by Yang et al. to verify whether our method can improve the robustness of the pre-trained purifier model. More importantly, this part is only discussed as the ablation study, and we put the robustness comparison experiment with other SOTA methods in Section 4.2.

---

> > ### Author Response · Authors · 2023-11-17
> >
> > We appreciate again for your effort review. Have our responses addressed your questions? And we are open to further discussion if you have any other concerns.

---

> > > ### Comment · Reviewer_onZy · 2023-11-22
> > > **Thanks for the rebuttal**
> > >
> > > Hello authors, thank you for the detailed answers. My concerns were mainly addressed. From my side, I am increasing my score for now. I am looking forward to read your responses to the other reviewers.

---

> > > > ### Author Response · Authors · 2023-11-22
> > > >
> > > > Dear Reviewer onZy,
> > > >
> > > > Thanks for your constructive feedback and recognition of our work. We are pleased to hear that your main concerns have been addressed and appreciate your increased score. We will promptly address the comments of the other reviewers in an effort to improve the quality of our paper further. Thank you again for your valuable time and effort.

---

> > ### Comment · Reviewer_jVyX · 2023-11-21
> > **Transferability of the Purification Models**
> >
> > > A2: One of the advantages of AP is that once the generator model is fully trained, it will be a plug-and-play module that can be applied to other model architectures.
> >
> > Since the adversarial training of the purification model $g$ relies on a specific classification model $f$, it might be the case that after AToP, $g$ is tailored for $f$, and applying $g$ to another classification model $f'$ may not yield satisfactory results, e.g., using the pre-trained weights may be better than using the fine-tuned weights regarding $f$. Hence, there should be some experimental results to support this claim.

---

> > > ### Author Response · Authors · 2023-11-21
> > >
> > > Thank Reviewer jVyX for your comments.
> > >
> > > Firstly, we clarify our understanding of Reviewer onZy's question: Whether AToP can perform well on various architectures. In response, the purifier model is a plug-and-play module, so our method can be integrated into any architecture.
> > >
> > > Secondly, we will address the new question you raised: whether the fine-tuned generator models will lead to 'overfitting' on the classifier used during training, potentially decreasing robust accuracy on other classifiers.
> > >
> > > Actually, in our paper, we conduct relevant experiments on this question. In each Table, including Table 2, 3, and 4, the purifier model is trained under the same classifier and tested on two different classifiers. For instance, in Table 4, we fine-tune the purifier model within WideResNet-28-10, and the final purifier model is tested on WideResNet-28-10 and WideResNet-70-16. The results show that the purifier model performs very well on other classifiers (WideResNet-70-16). Here is our understanding of the results: As mentioned in [1] and [2], 'One of the most intriguing properties of adversarial examples is that they transfer across models with different architectures and independently sampled training sets. Given that the adversarial perturbation features are inherent to the data distribution, different classifiers trained on independent samples from that distribution are likely to utilize similar non-robust features.' Based on the existing results, when fine-tuning the purifier with classification loss on one classifier, the model also shows transferability on different classifiers. We preliminarily think that similar to the principles concerning the aforementioned attacks, the defense may also exhibit transferability across classifiers.
> > >
> > > Although we conducted experiments in the paper, we did not discuss them in the paper. Therefore, we appreciate the comments you have raised. We think this is a point that is indeed worthy of discussion. We will add further analysis into the camera-ready stage.
> > >
> > > [1] Adversarial Examples Are Not Bugs, They Are Features. 2019.
> > >
> > > [2] Feature Purification: How Adversarial Training Performs Robust Deep Learning. 2020.

---

### Official Review · Reviewer_n69U · 2023-10-30

**Soundness:** 2 fair
**Presentation:** 3 good
**Contribution:** 2 fair
**Rating:** 5
**Confidence:** 5

**Summary:**

1. This paper aims to enhance the adversarial robustness of deep models.

2. Combining adversarial training and purification techniques, a new pipeline is proposed, i.e., adversarial training on purification.

3. Different from purification methods that purify adversarial examples with a pre-trained generative model before classification,
the generative model can be finetuned by adversarial training in the proposed pipeline. Moreover, the introduce random transformations play an important role in defensing adversarial attacks.

**Strengths:**

1. The paper is written clearly and easy to follow.
2. Models trained by the proposed method show promising robustness and generalization.

**Weaknesses:**

1. What if there are no random transformations in the proposed pipeline?
    Does the robustness come from the gradient vanishing from random operations?

2. For adversarial training methods, auto-attack is one of the most effective attack methods to evaluate model robustness. This is because models with adversarial training usually show much better results under FGSM, PGD, and other attacks than under the auto-attack.

    However, as indicated by Table 7 and Table 8 in this paper, the trained model even achieves much better robustness under auto-attack than under PGD20, FGSM attacks.
    From this perspective, the comparisons in Tables 2,3,4,5 can be unfair because models with adversarial training show worst-case robustness under auto-attack while it is not the case for the model trained by the proposed method.

3. Considering the issue in 2, to demonstrate their robustness, the models trained by the proposed methods should be fully tested under different attacks. For example, the CW attack with a large number of steps, and the PGD attack with 1000 steps.

**Questions:**

See above weaknesses.

**Details Of Ethics Concerns:**

No ethics concerns.

---

> ### Author Response · Authors · 2023-11-13
>
> We greatly appreciate the comments of the Reviewer n69U. Below are our responses to the questions raised.
>
> A1.1: If there are no random transformations, the robustness will decrease. $RT_2$ adds more randomness based on $RT_1$, and its robust accuracy has been improved. At the same time, Cohen and Wu et al. theoretically provide robustness guarantees about randomized smoothing, which is described in Section 3.1.
>
> A1.2: Random operations contribute to robustness but not fully due to gradient vanishing. As mentioned in A1.1, there are some theories of robustness guarantees about random operations. Indeed, gradient vanishing does provide 'illusory' robustness for some attacks. Regarding this issue, we also validate our method with AutoAttack and BPDA.
>
> A2: We conduct AutoAttack verification under two different settings: In Table 2,3,4,5, we attack the entire (purifier+classifier) model, and in Table 7 and 8, we attack the classifier model. The result in Table 7 and 8 is reasonable, but we also realize that the current tables can easily lead to misunderstandings. Therefore, we plan to redesign the Table 7 and 8 according to your comments, replacing AutoAttack with CW and PDG-1000.
>
> Here is the explanation of the results: Considering the robustness misestimation caused by obfuscated gradients of the purifier model, we use BPDA (Athalye et al., 2018a) and follow the setting by Yang et al. (2019), approximating the gradient of the purifier model as 1 during the backward pass, which is equivalent to attacking the classifier model. To maintain the same settings, AutoAttack only attacks the classifier. The advantage of AutoAttack lies in using EOT and adaptive mode with white-box and black-box attacks to attack more complex architectures, such as black-box and random factors in the architectures. If there are some random factors in the classifier, although the classifier model can effectively defend against FGSM and PGD, it can still be vulnerable to AutoAttack. However, in classic classifiers (such as Resnet-18), these factors do not exist, and simple white-box attacks cause more damage. Under standard cases, the entire model should be input into AutoAttack directly, and the experimental results are shown in Table 2,3,4,5. We will redesign Table 7 and 8 to avoid confusion.
>
> | Standard Case ($RT_2, \epsilon=8/255$) |  | | |
> |:---:|:---:|:---:|:---:|
> | Generator models | AToP | CIFAR-10 | CIFAR-100 |
> | AE (Wu et al., 2023) | $\times$ | 44.73 | 38.48 |
> | AE (Wu et al., 2023) | $\checkmark$ | **59.18** | **40.23** |
> | GAN (Ughini et al., 2022) | $\times$ | 59.57 | 38.89 |
> | GAN (Ughini et al., 2022) | $\checkmark$ | **72.85** | **43.55** |
>
>
> A3: Thanks for the comments. The following are the additional experiments on Table 7 and 8:
>
> | | | CIFAR-10 | | ImageNette | |
> |:---:|:---:|:---:|:---:|:---:|:---:|
> | Transforms | AToP | CW-100 | PGD-1000 | CW-100 | PGD-1000 |
> | RT1 | $\times$ | 21.09 | 0.00 | 64.26 | 0.00 |
> | RT1 | $\checkmark$ | **87.89** | **39.45** | **78.12** | **47.07** |
> | RT2 | $\times$ | 87.70 | 67.97 | 86.52 | 77.34 |
> | RT2 | $\checkmark$ | **88.87** | **84.38** | **88.48** | **82.03** |
> | RT3 | $\times$ | 75.20 | 70.31 | 67.38 | 66.21 |
> | RT3 | $\checkmark$ | **76.56** | **73.44** | **78.32** | **72.85** |

---

> > ### Author Response · Authors · 2023-11-17
> >
> > We appreciate again for your effort review. Have our responses addressed your questions? And we are open to further discussion if you have any other concerns.

---

> ### Comment · Reviewer_n69U · 2023-11-21
> **Random transformations for robustness**
>
> Thanks for your kind response.
>
> I still have some concerns.
> I agree that adding Gaussian noise is helpful to robustness, which has been identified by Li et al. with theoretical proof.
> However, why does the random masking scheme lead to significant robustness improvements?
> If the improvements don't come from obfuscated gradients, could the authors give some explanations for it?

---

> > ### Author Response · Authors · 2023-11-21
> >
> > Thanks for the further comments.
> >
> > The random masking does not provide robustness.
> >
> > We think there has been some misunderstanding. Allow us to clarify once again: In the paper, random transform (RT) principally comprises two components: random smoothing (adding Gaussian noise) and random masking. Random smoothing is helpful to robustness. As mentioned, this point is supported by both experimental and theoretical evidence. When only using random masking ($RT_1$), it does not provide robustness. As shown in the table below (the key sections from Table 7 and 8), $RT_1$ can hardly defend against attacks. Therefore, the attack is effective, and the model did not result in 'illusory' robustness due to the obfuscated gradients brought by random masking.
> >
> > | | | | | | |
> > |:---:|:---:|:---:|:---:|:---:|:---:|
> > | Table 7 | | | | | |
> > | Transforms | AToP | Standard Acc. | FGSM | PGD-10 | PGD-20 |
> > | RT1 | $\times$ | 93.36 | 16.60 | 0.00 | 0.00 |
> > | Table 8 | | | | | |
> > | Transforms | AToP | Standard Acc. | FGSM | PGD-10 | PGD-20 |
> > | RT1 | $\times$ | 91.99 | 18.95 | 0.00 | 0.00 |
> >
> > At the same time, random masking is not the contribution of our paper. Our contribution lies in proposing a new defense technique (AToP), which can be applied to various generator-based purifier models. We test AToP on two of the latest models (Ughini et al., (2022) and Wu et al., (2023)). Since both are based on the image inpainting pipeline, we use random masking as the default processing.

---

> > > ### Comment · Reviewer_jVyX · 2023-11-22
> > > **Discussion on the Contribution of Random Masking to Robustness**
> > >
> > > The comparison between RT1 and RT2 does suggest that random masking alone does not produce robustness. However, it is possible that the randomness of the masking partially contributes to the robustness of the models with RT2.
> > >
> > > I have a suggestion for investigating the contribution of the randomness of masking to the robustness: to consider a variant of RT2 that fixes the random mask $m$ to be a constant $m_0$, i.e., $m=m_0$ for each forward of the model. The comparison between RT2 and this variant should provide extra insights into how different components of the model contribute to the robustness.

---

> > > > ### Author Response · Authors · 2023-11-22
> > > >
> > > > Thank Reviewer jVyX for discussing the contribution of random masking to robustness.
> > > >
> > > > Your comment is helpful. However, thoroughly demonstrating the impact of random masking requires retraining the purifier model with a fixed mask $m$, which will consume a significant amount of time. At present, we can only offer some of our understanding: We think that a fixed mask $m$ may result in lower robust accuracy, as randomness is a good trick in enhancing robust accuracy. There are two points that can easily be confused here: certified robustness and empirical-based robust accuracy. As mentioned in existing studies [1], although randomness as a trick can enhance robust accuracy, it may not essentially improve the model robustness.
> > > >
> > > > We will retrain a purifier based on a fixed mask $m$ and add these experimental results and discussion into the appendix. However, we still want to claim that the robustness of random masking is not our contribution. Our contribution lies in proposing a new defense technique (AToP), which can be applied to various generator-based purifier models and enhance robust accuracy.

---

> > > > > ### Comment · Reviewer_jVyX · 2023-11-22
> > > > > **Discussion on the Contribution of Random Masking to Robustness (cont.)**
> > > > >
> > > > > Thanks for your comment. I think fixing the mask $m$ **solely at test time** and evaluating the existing trained models suffices to prove that the randomness of the mask plays a role in improving the robustness (if that is the case).
> > > > >
> > > > > I understand that random masking is not the contribution of this paper, and I'm just curious whether this randomness matters.

---

> > > > > > ### Author Response · Authors · 2023-11-22
> > > > > >
> > > > > > Thanks for your prompt response and understanding. To satisfy your curiosity, we tested RT2 under the conditions you suggested. The experimental results are shown below. When using a fixed mask $m$, the robust accuracy has decreased.
> > > > > >
> > > > > > | PGD-10 ($\epsilon=8/255$) |  | | |
> > > > > > |:---:|:---:|:---:|:---:|
> > > > > > | Transforms | AToP | Random mask | Fixed mask |
> > > > > > | RT2 | $\times$ | 72.27 | 70.12 |
> > > > > > | RT2 | $\checkmark$ | **84.77** | **79.10** |

---

### Official Review · Reviewer_jVyX · 2023-11-01

**Soundness:** 2 fair
**Presentation:** 2 fair
**Contribution:** 3 good
**Rating:** 6
**Confidence:** 4

**Summary:**

This paper proposes Adversarial Training on Purification (AToP) for adversarial defense, which is expected to improve the robustness while achieving good generalization ability. The proposed method consists of two components: perturbation destruction by random transformations and a purifier model that aims to recover the clean image which is adversarially finetuned. The experiments suggest that AToP can improve the robustness against both seen and unseen attacks without significantly sacrificing the accuracy on clean images, and achieve state-of-the-art results on several benchmarks.

**Strengths:**

- This paper makes an early attempt to finetune an adversarial purification model with adversarial training, and suggests that this can be beneficial.
- The experiments are complete and the results seem promising.
- The figures are clear and meaningful.

**Weaknesses:**

- Clarity of Section 3.2.
  - Eq. (8) seems to be a mix of the loss functions for the generator (purifier) $g$ and the discriminator $D$, which are supposed to be different in GAN training. Besides, since AToP is not limited to GAN-based purifier models, it is better to present a general form of the loss function.
  - It is stated that "the discriminator $D$ is responsible for distinguishing between real examples (*including clean or adversarial examples*) and the purified examples", but this is not reflected by Eq. (8) since only the clean sample $x$ is involved in $L_{df}$.
  - Line 5 of Algorithm 1 is confusing. The notation $g_{\theta_g}$ is not explained.
- Some experimental details are not clearly stated.
  - For evaluation, it is claimed that "we randomly select 512 images from the test set for robust evaluation." It is not clear whether this applies to all results or all compared methods. The comparison may be unfair if different models are evaluated on different test sets.
  - The "Ours" in Table 2,3,4 are not specified: Which RT and purifier model are used? What are the hyper-parameters for the training attack?
- The results for AToP in Table 8 may be unreasonable. Specifically, considering the accuracy of the same model against different attacks, it is shown that FGSM < PGD < AutoAttack, which is exactly the reverse of common expectations since AutoAttack is believed to be a much stronger attack while FGSM should be the weakest attack. These results may significantly challenge the reliability of the robustness evaluation and should be seriously discussed.

**Questions:**

- Why FGSM is used for adversarial finetuning instead of PGD or other attacks?

- Can the reconstruction loss in Eq. (8) be applied to the reconstruction of adversarial images? Specifically, is it likely that adding $\mathbb{E}[\Vert x-g(t(x+\delta),\theta_g) \Vert]$ to the loss will increase the performance?

- Why does reconstruction loss take $\ell_1$-distance?

- What is the expected scope of purification models that AToP can generalize to?

---

> ### Author Response · Authors · 2023-11-13
>
> We greatly appreciate the comments of the Reviewer jVyX. Below are our responses to the questions raised. Due to exceeding the character limit for a single reply, we divide the response into two parts (1 / 2).
>
> The first part is about weaknesses:
>
> A1.1: We also realize this might be confusing, so we have revised Eq.(8) in A1.2. Regarding the second point, we agree with your perspective. We have only used the GAN-based model as an illustrative example. We will add a general form: $L_{\theta_{g}} = L_{org}\left(x, \theta_{g}\right)+\lambda \cdot L_{c l s}\left(x, y, \theta_{g}, \theta_{f}\right)$. For instance, the original loss function is denoted as $L_{df}$ in the GAN-based model (Ughini et al., 2022), and as $L_{mae}$ in the MAE-based model (Wu et al., 2023).
>
> A1.2: Thanks for the comments. Eq.(8) should be divided into two stages, including pre-training and fine-tuning:
>
> Pre-training:
>
> $L_{\theta_{g}} =L_{df}(x, \theta_{g})$
>
> $=\mathbb{E}[D(x)]-\mathbb{E}[D(g(t(x),\theta_g))]+\mathbb{E}[{\|x-g(t(x), \theta_g)\|}_{\ell_1}],$
>
> Fine-tuning:
>
> $L_{\theta_{g}} =L_{df}(x', \theta_{g})+\lambda \cdot L_{c l s}(x', y, \theta_{g}, \theta_{f})$
>
> $ =\mathbb{E}[D(x')]-\mathbb{E}[D(g(t(x'),\theta_g))]+\mathbb{E}[{\|x'-g(t(x'), \theta_g)\|}_{\ell_1}]$
>
> $ +\lambda \cdot \max_{\delta} CE(y, f(g(t(x'),\theta_g))), \quad where \ x' = x+\delta.$
>
> A1.3: $g_{\theta_g}$ represents the computed gradient (i.e., grad\_$\theta_g$). We will change the $g_{\theta_g}$ in the Algorithm 1 to $\nabla \theta_g$ to avoid confusion.
>
> A2.1: Due to the high computational cost of testing models with multiple attacks, we conducted experiments based on the guidance provided by Nie et al. In fact, not all experiments tested 512 images. However, Nie et al. have demonstrated through experiments (Appendix C.2 in [1]) that the robust accuracies of most baselines do not change much on the sampled subset, compared to the whole test set.
>
> [1] Diffusion Models for Adversarial Purification. 2022.
>
> A2.2: We will add this part of the information. We utilize $RT_2$ and GAN-based models, which yielded the best results in the experiment. The adversarial examples are generated by the FGSM during training.
>
> A3: The experimental results in this section are correct, and similar results (FGSM < PGD, Table 1 in [2]) are observed in the work of Ughini et al. And we conduct AutoAttack verification under two different settings: In Table 2,3,4,5, we attack the entire (purifier+classifier) model, and in Table 7 and 8, we attack the classifier model. We also realize that the current tables can easily lead to misunderstandings. Therefore, we plan to redesign the Table 7 and 8 according to Reviewer n69U's comments, replacing AutoAttack with CW and PDG-1000.
>
> Here is the explanation of the results: Considering the robustness misestimation caused by obfuscated gradients of the purifier model, we use BPDA (Athalye et al., 2018a) and follow the setting by Yang et al. (2019), approximating the gradient of the purifier model as 1 during the backward pass, which is equivalent to attacking the classifier model. To maintain the same settings, AutoAttack only attacks the classifier. The advantage of AutoAttack lies in using EOT and adaptive mode with white-box and black-box attacks to attack more complex architectures, such as black-box and random factors in the architectures. If there are some random factors in the classifier, although the classifier model can effectively defend against FGSM and PGD, it can still be vulnerable to AutoAttack. However, in classic classifiers (such as Resnet-18), these factors do not exist, and simple white-box attacks cause more damage. Under standard cases, the entire model should be input into AutoAttack directly, and the experimental results are shown in Table 2,3,4,5. We will redesign Table 7 and 8 to avoid confusion.
>
> [2] Trust-No-Pixel: A Remarkably Simple Defense against Adversarial Attacks Based on Massive Inpainting. 2022.

---

> > ### Comment · Reviewer_jVyX · 2023-11-15
> > **Doubts About the Evaluation Protocol for Ablation Studies**
> >
> > Thanks for your valuable response. Most of my concerns have been addressed, but I still have some doubts about the evaluation protocol for Section 4.4 (Table 7,8). Given that only the classifier is involved in the attack, the results suggesting that FGSM is more effective are indeed reasonable. Nonetheless, **why is the evaluation protocol in Section 4.4 different from that in Section 4.1**? Considering that attacking the classifier only is not an effective approach, the evaluation protocol of Section 4.1 should be applied to all experiments in my opinion.

---

> > > ### Author Response · Authors · 2023-11-20
> > >
> > > We appreciate again for the insightful comments and hope your concerns are adequately addressed. If you are satisfied with the rebuttal, we kindly request that you consider raising the score. If not, we would be happy to continue the discussion during the rebuttal period.

---

> ### Author Response · Authors · 2023-11-13
>
> We greatly appreciate the comments of the Reviewer jVyX. Below are our responses to the questions raised. Due to exceeding the character limit for a single reply, we divide the response into two parts (2 / 2).
>
> The second part is about questions:
>
> QA1: The training cost of using FGSM for adversarial fine-tuning is acceptable. At the same time, as we propose a new paradigm that considers both AT and AP, we use FGSM to verify our ideas preliminarily. Here is a simple experiment regarding time, where we have measured the time required to process 10,000 images:
>
> | Time costs |  |  |
> |-------------------|:-----------:|:-----------:|
> |  | FGSM   | PGD-10 |
> | GAN with AToP     | ~186 sec  | ~1469 sec |
>
> QA2: The impact of optimization using consistency loss of adversarial examples is limited. We can employ a variety of attacks and obtain a variety of different adversarial examples. When using consistency loss to learn information from adversarial examples, which potentially leads to overfitting and weakens the generalization ability against unseen attacks. We also discuss this point in the related work section. Additionally, as the perturbations are minimal, in many strong attacks, the consistency loss from adversarial examples is less than the consistency loss from generated examples ($\mathbb{E}[x-g(x_{adversarial})] < \mathbb{E}[x-g(x_{generated})]$). This also results in the consistency loss not playing a significant role during training.
>
> QA3: In the pipeline based on the image completion, the repair results of the boundary of the missing area are better than the middle area, and pixels with different distances from the filled edge should be assigned different weights. A common way is to use the distance between pixels (i.e., $l_1$ distance) for weight calculation. When repairing large missing areas, the $l_1$ loss will be more effective in improving repair quality.
>
> QA4: We hope AToP can be applied to any pre-trained generator-based purifier model. To this end, we conduct experiments on two common generator-based purifier models, including GAN-based and AE-based. At the same time, we also attempted to use diffusion-based purifier models. However, as mentioned in the limitations section, our method cannot easily fine-tune all models yet, such as some models with high computational costs.

---

> ### Author Response · Authors · 2023-11-15
>
> Thanks for the further comments. We conduct some additional experiments and provide corresponding explanations, hoping that these can address your concerns.
>
> Due to the existence of obfuscated gradients and random factors in adversarial purification, attacks such as FGSM and PGD cannot calculate the correct perturbation. At this point, the model might result in 'illusory' robustness. The results of robust accuracy against FGSM on CIFAR-10 are shown in the table below. **The robust accuracy may be unreliable when testing on FGSM and PGD with the evaluation protocol from Section 4.1.** Therefore, regarding these attacks, Athalye et al. proposed a more reliable evaluation protocol, which has been widely used in adversarial purification tasks (Yang et al. and Ughini et al.). We also follow the same setting to validate the robustness against FGSM and PGD.
>
> | Robust Acc. on FGSM |  | | |
> |:---:|:---:|:---:|:---:|
> | Evaluation protocol | $RT_1$ | $RT_2$ | $RT_3$ |
> | From Section 4.4 | 16.60 | 55.08 | 67.97 |
> | From Section 4.1 | 39.84 | 72.66 | 67.19 |
>
> As you correctly mentioned, the evaluation protocol in Section 4.4 should not be applied to AutoAttack. Here are the results under the standard cases (Section 4.1) of AutoAttack.
>
> | Standard Case ($RT_2, \epsilon=8/255$) |  | | |
> |:---:|:---:|:---:|:---:|
> | Generator models | AToP | CIFAR-10 | CIFAR-100 |
> | AE (Wu et al., 2023) | $\times$ | 44.73 | 38.48 |
> | AE (Wu et al., 2023) | $\checkmark$ | **59.18** | **40.23** |
> | GAN (Ughini et al., 2022) | $\times$ | 59.57 | 38.89 |
> | GAN (Ughini et al., 2022) | $\checkmark$ | **72.85** | **43.55** |

---

> > ### Comment · Reviewer_jVyX · 2023-11-21
> > **Further Concerns about the Evaluation Protocol**
> >
> > Thanks for your further replies. Regarding the evaluation protocol in Section 4.4, I think that **using the best protocol for each attack** should be the principle for reliable robustness evaluation, and the revised results for AutoAttack seem reasonable.
> >
> > Nonetheless, as suggested by [a], BPDA may not be the most effective way to apply PGD attacks. Instead, PGD with Expectation over Transformation (EOT) is shown to be much more effective than PGD+BPDA for DiffPure and other purification-based defenses, and it even results in lower robust accuracy than AutoAttack as reported in [a]. Therefore, it would be better to test the models with **PGD+EOT instead of PGD+BPDA** in the experiments.
> >
> >
> > [a] Lee, Minjong, and Dongwoo Kim. "Robust evaluation of diffusion-based adversarial purification." ICCV 2023.

---

> > > ### Author Response · Authors · 2023-11-21
> > >
> > > Thanks for the further comments.
> > >
> > > Due to the presence of obfuscated gradients, the experiments with BPDA are necessary. We are grateful for the paper you provided, and we agree that the experiments with EOT can better demonstrate the effectiveness of the method. Ultimately, we have decided to incorporate the EOT experiments while retaining the BPDA experiments. However, due to time constraints, we have only been able to obtain partial results, as shown in the table below. And we plan to adjust the remaining parts in camera-ready stage.
> > >
> > > | CIFAR-10 ($\epsilon=8/255$) |  | | |
> > > |:---:|:---:|:---:|:---:|
> > > | Transforms | AToP | Standard Acc. | PGD+EOT |
> > > | RT1 | $\times$ | **93.36** | 0.00 |
> > > | RT1 | $\checkmark$ | **93.36** | **43.16** |
> > > | RT2 | $\times$ | 84.18 | 69.14 |
> > > | RT2 | $\checkmark$ | **90.04** | **81.05** |
> > > | RT3 | $\times$ | 75.98 | 68.55 |
> > > | RT3 | $\checkmark$ | **80.02** | **72.07** |

---

> > > > ### Comment · Reviewer_jVyX · 2023-11-22
> > > > **Further Discussion about the EOT Attack**
> > > >
> > > > Thanks for the updated results of PGD+EOT, and I acknowledge that reporting both BPDA and EOT results is valuable.
> > > >
> > > > It seems that the robust accuracy of ResNet-18 with AToP (RT2) under PGD+EOT is surprisingly high, and I think the implementation of EOT should be clarified, since certain implementation designs may lead to unreliable evaluation of the robustness according to [a]. Specifically:
> > > > - How many EOT samples do you use for EOT attacks?
> > > > - Do you use any numerical approximation for the computation of gradients?
> > > > - Do you skip any components of the model in the gradient computation?
> > > >
> > > > Besides, do you plan to release the code and the models? This can be crucial for other researchers to validate the robustness of these models, which possibly reaches the state-of-the-art.
> > > >
> > > > [a] Lee, Minjong, and Dongwoo Kim. "Robust evaluation of diffusion-based adversarial purification." ICCV 2023.

---

> > > > > ### Author Response · Authors · 2023-11-22
> > > > >
> > > > > Thanks for your prompt response.
> > > > >
> > > > > Due to time constraints, the new results of EOT are preliminary experiments. In the experiments, we use PGD with 10 steps and set the number of EOT to 10. In [a], they use PGD with 200 steps and set the number of EOT to 20. On the other hand, [a] conducts a detailed evaluation specifically targeting the diffusion-based model, whereas our experiments focused on the GAN-based model. If we keep the same parameter settings in [a], it will significantly increase the time required. To expedite the presentation of experimental results, we conduct preliminary experiments using smaller parameters and numerical approximation for the computation of gradients. These series of differences lead to such outcomes. Here are some settings to answer your question:
> > > > >
> > > > > 1: We set the number of EOT to 10.
> > > > >
> > > > > 2&3: We follow the same pipeline in Table 7. We use numerical approximation for the computation of gradients and skip the purifier model part.
> > > > >
> > > > > We agree to follow the settings in [a]. Compared to PGD-10, PGD-10 + EOT exhibits a lower robust accuracy, indicating that EOT indeed serves as an effective evaluation metric.
> > > > >
> > > > > We are currently conducting detailed experiments on EOT. We will cite [a] for guiding our experiments and providing relevant analyses. This part will be added to the paper.
> > > > >
> > > > > We are enthusiastic about advancing the development of the open-source community. After paper acceptance, we will release the code.
> > > > >
> > > > > [a] Lee, Minjong, and Dongwoo Kim. "Robust evaluation of diffusion-based adversarial purification." ICCV 2023.

---

> > > > > > ### Comment · Reviewer_jVyX · 2023-11-22
> > > > > > **Further Discussion about the EOT Attack (cont.)**
> > > > > >
> > > > > > Thanks for your clarifications. Skipping the purifier model in the gradient computation may lead to significantly higher robust accuracy, and I hope that more reliable evaluations following the settings in [a] can be conducted. I understand that this can be difficult due to time constraints, so I suggest that you report the results on a smaller subset of the test samples (e.g., 64 samples) if appropriate.

---

> > > > > > > ### Author Response · Authors · 2023-11-22
> > > > > > >
> > > > > > > Thanks for your prompt response and understanding. We test on 64 samples without 'skipping' and follow the same settings in [a]. The results are shown below:
> > > > > > >
> > > > > > > | CIFAR-10 |  | | |
> > > > > > > |:---:|:---:|:---:|:---:|
> > > > > > > | Transforms | AToP | Standard Acc. | PGD+EOT |
> > > > > > > | RT2 | $\times$ | 84.18 | 56.25 |
> > > > > > > | RT2 | $\checkmark$ | **90.04** | **64.06** |

---

> > > > > > > > ### Comment · Reviewer_jVyX · 2023-11-23
> > > > > > > > **Summary of the Discussion**
> > > > > > > >
> > > > > > > > I sincerely thank the authors for their responses and I appreciate their active participation and considerable efforts during the discussion period.
> > > > > > > >
> > > > > > > > My major concern with this paper is the robustness evaluation of the proposed models. The latest preliminary results provided by the authors with PGD+EOT applied to the whole model seem reasonable and suggest that the proposed method is indeed effective, possibly reaching state-of-the-art, although further comprehensive experiments with this evaluation protocol are necessary for drawing reliable conclusions.
> > > > > > > >
> > > > > > > > In my opinion, this paper makes a valuable contribution to the field of purification-based adversarial defense, but the following points are required for this paper to be above the acceptance level:
> > > > > > > > - Improve the clarification of the text, especially the Method section.
> > > > > > > > - Evaluate the models using PGD+EOT as discussed, apart from AutoAttack and BPDA. To compare with existing models using this same evaluation protocol is preferable.
> > > > > > > > - Release the code and the trained models so that other researchers can validate the robustness.
> > > > > > > >
> > > > > > > > Overall, considering the existing state of the paper after discussion and the promises of the authors, I decided to increase my rating from 5 to 6 for now. I hope that the authors can improve the paper accordingly.

---

> > > > > > > > > ### Author Response · Authors · 2023-11-23
> > > > > > > > >
> > > > > > > > > Dear Reviewer jVyX,
> > > > > > > > >
> > > > > > > > > Thank you for your constructive feedback and recognition of our work. We deeply appreciate your time and efforts in reviewing and actively discussing our paper.
> > > > > > > > >
> > > > > > > > > We understand and agree with your comments about our method. We will clarify and discuss all the concerns in the paper.
> > > > > > > > >
> > > > > > > > > 1: Thank you for your suggestions on clarification, which are very helpful to us. We are currently optimizing the paper to further improve readability.
> > > > > > > > >
> > > > > > > > > 2: We are using PGD+EOT for detailed comparative experiments with the same evaluation protocol, and more experiments will be added to the paper.
> > > > > > > > >
> > > > > > > > > 3: After paper acceptance, we will release the code and the trained models to the public, which are easier for researchers to use.
> > > > > > > > >
> > > > > > > > > We are pleased to hear that your main concerns have been addressed, and your increased score is much appreciated. Thank you again for your precious time and efforts.

---

### Comment · Area_Chair_G73w · 2023-11-20
**Comments on Authors' responses**

Dear Reviewers, The authors have responded to your valuable comments. Please take a look at their responses! Thanks!

---

### Author Response · Authors · 2023-11-21

Dear Reviewers,

We have submitted the latest version of the paper and hope your concerns are adequately addressed. If you are satisfied with this version, we kindly request that you consider raising the score. If not, we would be happy to continue the discussion during the rebuttal. Although the rebuttal period is coming to a close, we still hope to receive any responses and will do our best to answer.

---

> ### Author Response · Authors · 2023-11-22
> **Replying to the Public Comment**
>
> Replying to the Public Comment (**Update:  Since the person who responded has deleted the comments, please ignore the following response**):
>
> Thanks for your interest in our work! Below are our responses to the concerns raised.
>
> 1: Since this paper [0] has not been formally published, we are unaware of this work. We appreciate you sharing it with us.
>
> After reading, we think there is a significant difference between our work and [0]. Firstly, the problems we focus on are different: The current mainstream SOTA method is based on the pre-trained generator model. Therefore, we consider the limitation of the existing pre-trained generator-based purifier model, where all parameters are fixed and cannot be further improved for known attacks, rather than simply proposing a more robust AE model or GAN model. Secondly, due to the difference in problems, there are differences in our algorithms as well. In [0], both clean examples and adversarial examples are input simultaneously to optimize the AE model. Our method comprises two stages: the pre-training stage with clean examples and the fine-tuning stage with adversarial examples. This makes our method more widely applicable rather than being constrained to a specific generator-based purifier model. Therefore, although there is a similarity in the high-level concepts, the problems and specific algorithms are different. At the same time, we are addressing challenges in the most recent methods.
>
> 2: In our evaluation experiments, we have utilized EOT, as mentioned in Section 4.1: *'... and utilizes Expectation Over Time (EOT) (Athalye et al., 2018b) to tackle the stochasticity introduced by random transforms.'*
>
> 3: In the paper, we do not conduct any robustness evaluation experiments on the diffusion-based model. But we guess your 2nd and 3rd points may be that you hope we can conduct experiments using PGD+EOT. On this point, we have also had discussions with the Reviewer jVyX. We are currently conducting detailed experiments on EOT. This part will be updated in the paper. And you can find more details in our conversation.
>
> 4: We are confused about the fourth point. Why do you say *'the GAN-based model is notably weak, enhancement is relatively easy'*? Are there any studies about this viewpoint? At the same time, we test AToP on two of the latest models (Ughini et al., (2022) and Wu et al., (2023)).

---

> > ### Public Comment · ~LiuYiming1 · 2023-11-24
> >
> > Thank you for your generous reply, and since you added the results of accurate gradient backpropagation based on the reviewer jVyX’s suggestion, I think my main concerns have been dispelled, and I really hope you with a good result.
> > I support your method (ATOP) should be effective, especially generative models combined with Gaussian noise in fig.2(b).
> >
> > Here are some suggections to make your work stronger:
> > You can follow up with additional results based on more samples and EOT rounds,the accurate gradient calculation of GANs may not too difficult.
> > The settings used in the experiment can be stated more clearly to avoid misunderstandings.

---

### Meta-Review · Area_Chair_G73w · 2023-12-05

**Metareview:**

In this paper, the authors proposed Adversarial Training on Purification (AToP) that is composed of AT and purification to improve robustness and generalization.
There are several rounds of active discussions between the authors and reviewers.
Almost comments from the reviewers have been well addressed in that the authors clarify their technical details and contributions, provide more experimental results per reviewers' request.
Most importantly, all reviewers consistently reach a consensus of recognizing the merits of AToP.
Therefore, the AC suggests to accept this paper.

**Justification For Why Not Higher Score:**

none

**Justification For Why Not Lower Score:**

The authors' responses mostly satisfy the reviewers.

---

### Decision · Program_Chairs · 2024-01-16

Accept (poster)